# Research Progress on Mechanical Strength of Rice Stalks

**DOI:** 10.3390/plants13131726

**Published:** 2024-06-22

**Authors:** Huimin Yang, Jiahui Huang, Yuhan Ye, Yuqing Xu, Yao Xiao, Ziying Chen, Xinyu Li, Yingying Ma, Tao Lu, Yuchun Rao

**Affiliations:** College of Life Sciences, Zhejiang Normal University, Jinhua 321004, China; 19519802611@163.com (H.Y.); jhhuang08@163.com (J.H.); yeyuhan70@gmail.com (Y.Y.); yq9770423@163.com (Y.X.); 19557879740@163.com (Y.X.); c2984304896@163.com (Z.C.); 15541633805@163.com (X.L.); mayingying767@163.com (Y.M.); lutao980723@163.com (T.L.)

**Keywords:** rice, lodging resistance, mechanical strength of the stalk, cellulose, lignin, hemicellulose

## Abstract

As one of the most important food crops in the world, rice yield is directly related to national food security. Lodging is one of the most important factors restricting rice production, and the cultivation of rice varieties with lodging resistance is of great significance in rice breeding. The lodging resistance of rice is directly related to the mechanical strength of the stalks. In this paper, we reviewed the cell wall structure, its components, and its genetic regulatory mechanism, which improved the regulatory network of rice stalk mechanical strength. Meanwhile, we analyzed the new progress in genetic breeding and put forward some scientific problems that need to be solved in this field in order to provide theoretical support for the improvement and application of rice breeding.

## 1. Introduction

Rice (*Oryza sativa* L.), one of the world’s most important food crops, occupies an important position in agricultural production. Improving rice yield is one of the most important initiatives for ensuring food security. However, lodging is the key factor restricting rice yield, mostly occurring in the late grain filling stage, when photosynthetic products and nutrients stored in rice stalks and leaf sheaths are transferred to the grains, resulting in a decline in the mechanical strength of the stalks and a tendency towards lodging [1]. The occurrence of lodging seriously affects the development and harvesting of grains. And cultivating collapse-resistant rice has become an urgent need in current agricultural production.

However, the genetic indicators of lodging resistance traits in rice are still unclear. Since the 1960s, dwarf breeding in the “Green Revolution” has played a key role in solving the problem of rice lodging [2,3]. However, it is difficult to increase rice yields due to the reduction in plant length followed by the reduction in production. Lin Zechuan et al. found that reducing the plant tiller angle appropriately can improve the resistance of rice to collapse [4]. Liu Chang et al. found that plant spike type was also related to the lodging resistance of rice and that upright spike varieties were more resistant to lodging than curved spike varieties [1]. Nonetheless, a single improvement to these traits is not a good solution to the problem of lodging in agricultural production. The improvement of the rice plant’s resistance to lodging, whether from the plant height, tillering, or spike standpoint, presupposes that the mechanical strength of the plant’s stalks is sufficient to support the entire weight of the plant. Therefore, stem mechanical strength is feasible as an effective indicator from which to study the genetic regulation mechanism of the resistance to lodging [1,5,6]. Currently, many scholars have suggested that stem mechanical strength is significantly and positively correlated with plant resistance to lodging [6,7,8], and Kashiwagi et al. found that plant height and yield were less affected when stem mechanical strength was used as an indicator to improve plant resistance to lodging [7]. Therefore, analyzing the molecular regulatory mechanism of rice lodging resistance by using plant stem mechanical strength as an indicator can effectively expand the possibility space in improving rice lodging resistance, which has very broad prospects in application.

## 2. Cell Wall of Rice Stalks

### 2.1. Cell Wall Structure of Rice Stalks

A transverse observation of rice stalk tissue reveals that the cell wall structure mainly consists of a primary cell wall and a secondary cell wall. In particular, the primary cell wall contains cellulose, hemicellulose, pectin, enzymes, and structural proteins, and its basic structure is a polysaccharide cross-linked skeleton [9]. Generally speaking, in the region of the vigorous cell division and growth of plant cells, the cell walls usually have only primary walls, and such cells are thin-walled cells, which can expand along with the growth of plant cells [10]. When cell growth ceases, some specialized cells, such as fibroblasts and xylem duct cells, thicken on the inside of the primary wall to form a secondary wall. In addition to cellulose and hemicellulose, the secondary cell wall also contains lignin, a small amount of pectin, protein, and aromatic compounds, and the cells containing secondary walls are thick-walled cells. Studies have shown that plant mechanical tissues play a supportive and protective role for the plant, and several layers of thick-walled cells located around the vascular tissue and below the epidermal layer play a major supporting role; these are the main structural factors affecting the mechanical strength of rice stems [11].

### 2.2. Cell Wall Components of Rice Stalks

The composition of the cell wall is an important factor affecting the structure and function of the cell wall. As the plant skeleton, the cell wall is mainly composed of cellulose, lignin, hemicellulose, and pectin [12]. The content of these components closely affects the mechanical strength of the plant stem.

#### 2.2.1. Cellulose

Cellulose is the most abundant polysaccharide in the primary walls and secondary walls of plants, and it participates as part of the basic skeleton constituting the cell wall network structure. There is a high cellulose content in mature rice stalks, accounting for about 1/3 of the dry weight of the plant, and it plays an irreplaceable role in regulating the mechanical strength of the stalks [13]. Wang Qingyan et al. used a rice dwarf mutant to study the relationship between cell wall components and mechanical properties and found that rice stalk strength was significantly and positively correlated with cellulose content [14,15]. Guo Yuhua et al. pointed out that the higher the density of cellulose within the stalk, the greater its mechanical support and the greater its resistance to lodging [16]. Thus, in-depth study of the structural and functional properties of cellulose is crucial for analyzing the regulation mechanism of rice stalk mechanical strength and cultivating lodging-resistant rice varieties with excellent traits.

Cellulose is an unbranched polysaccharide polymer molecule linked by β-1,4 glycosidic bonds. Its basic unit, the microfilament, is about 2.4–3.6 nm in diameter, rope-like, and contains about 36 glucan chains connected by hydrogen bonds and van der Waals forces. Cellulose is a high-level structure made up of numerous microfilaments that are further aggregated. In nature, Cellulose generally exists as crystalline cellulose, whereas cellulose in the primary wall is thought to be an ordered arrangement of crystallized and amorphous cellulose [17,18]. Crystallinity (CrI) and degree of polymerization (DP) are two important structural indicators of cellulose. Crystallinity is an average concept indicating the percentage of crystalline cellulose in total cellulose and can be determined via X-diffraction, whereas the degree of polymerization refers to the number of molecules in the dextran chain and can be measured via viscometry and gel permeation chromatography [19]. At present, there is still no unified understanding of the higher-order structure of cellulose.

#### 2.2.2. Lignin

Lignin is a class of aromatic polymers that are widely present in plant cell walls and are closely related to the mechanical strength of the stalk [20], the waterproofness of the cells, and the resistance of the plant to abiotic aggression [21]. Lignin monomers are direct precursors that make up lignin, including coumaryl alcohol, coniferyl alcohol, and sinapyl alcohol, which are produced from phenylpropane derivatives via hydroxylation, methylation, and a series of reduction reactions [22]. These monomers are, in turn, linked by ether and C-C bonds to further form three different lignin macromolecules: P-hydroxyphenyl lignin (H-lignin), lilac-based lignin (S-lignin), and guaiac-based lignin (G-lignin) [23]. The composition of lignin monomers varies in proportion in different plant species and is therefore also considered a marker of plant evolution.

In recent years, the role of lignin in maintaining the toughness of the cell wall and improving the plant’s resistance to lodging has received much attention from researchers. During lignification, lignin enters the cell wall skeleton—composed of macromolecules such as cellulose—through osmosis and binds to it, resulting in a stronger cell wall structure and improved mechanical strength [24]. From a cytological point of view, the enzymes PAL and CAD are key enzymes in the lignin synthesis pathway, and they also concurrently promote the differentiation of duct molecules, thereby enhancing stem toughness. Similarly, the deletion of CAD and CCR resulted in a significant reduction in Arabidopsis lignin content and changes in lignin composition, which will ultimately lead to severe inhibition of plant growth [25]. Li et al. used a variety of Monocotyledons and Dicotyledons as study materials to inhibit the expression of genes related to lignin synthesis and found that cell wall collapse was prevalent in these plants, which further revealed the relationship between the compressive strength of the plant cell wall and lignin content [26]. In addition, the interactions between lignin and other components have been widely reported. Years of research have shown that lignin and hemicellulose form complexes through the cross-linking of “hemicellulose—ester—ferulic acid—ether—lignin bridging”, which fill the cell wall gaps and participate in the formation of complex cell wall networks [27]. However, the direct mechanism of lignin and hemicellulose on cell wall composition has not yet been clarified, and the interactions with cellulose, pectin, etc., need to be further studied.

#### 2.2.3. Hemicellulose

Hemicellulose is the second most abundant polysaccharide in plant cell wall components after cellulose, and its main chain is also linked by a single β-1,4 glycosidic bond. Hemicellulose varies greatly in plants and can be classified as xylan (β-1,4-xylanoside in the main chain), xyloglucan (β-1,4-glucanoside in the main chain), mannan (β-1,4-mannanoside in the main chain), and mixed-linkage glucan (MLG, with main chains of β-1,4- and β-1,3-glucosides) according to the type of main-chain monosaccharides [28]. Among them, MLG and others contain branched chains, which are mainly composed of the neutral sugars xylose, galactose, and fucose. The predominant hemicellulose in the primary cell wall is xyloglucan, whereas the predominant hemicellulose in the secondary cell wall is xylan [29]. Within the cell wall, hemicellulose cross-links with cellulose to form matrix polysaccharides, while the arabinose of hemicellulose can increase cellulose crystallinity by cross-linking with cellulose β-1,4-glucan chains to enhance the mechanical strength of the rice stem, resulting in increased plant resistance to lodging.

#### 2.2.4. Other Components

In addition to cellulose, lignin, and hemicellulose, the cell wall components also include pectin and a small amount of protein [30].

Pectin, an acidic polysaccharide rich in galacturonic acid, is the most structurally complex polymer in the cell wall and is mainly composed of three structural domains: type I rhamnogalacturonan RGI, type II rhamnogalacturonan RGII, and homogalacturonan HG. They are connected by covalent bonds and play an important role in cell adhesion [30].

Plant cell wall proteins are mainly hydroxyproline-enriched proteins, including extensins, arabinogalactan proteins, lectins, and expansins [31]. Among them, extensins and expansins were discovered earlier, and their mechanisms of action were more comprehensively revealed. In the presence of peroxidase, extensins can spontaneously form insoluble gelatinous structures and form condensates with pectins, inducing the deposition of cell wall material at the cell plate [32]; whereas expansins are a class of cell-wall-specific enzymatic proteins, which are capable of disrupting the hydrogen bonding between cell wall polymers, resulting in cell wall relaxation [33].

### 2.3. Cell Wall of Rice Stalks and Mechanical Strength

It has been shown that plant cells form different types of cell walls at different developmental stages or in adverse environments, and these different types of cell walls have different mechanical properties [30]. In cells with high water pressure (e.g., xylem cells), the cell wall has high mechanical strength and can resist the squeezing force of water pressure. In contrast, in cells that require greater plasticity, such as mesophyll cells, the cell wall is softer to accommodate the various deformation needs of the cell. These differences all reflect the mechanisms by which the chemical composition and structure of the cell wall regulate the mechanical strength of the cell.

In recent years, researchers have used techniques such as genetic engineering and molecular biology to alter the expression of genes involved in the synthesis of plant cell walls to study the relationship between cell walls and mechanical strength. By up-regulating or down-regulating the expression of specific genes, the composition and structure of the cell wall can be altered, thereby affecting the mechanical strength of the cell. These studies not only revealed the regulatory network of cell wall synthesis but also provided new ideas for improving plant mechanical properties.

## 3. Molecular Regulatory Mechanisms of Major Components of Rice Stalk Cell Wall

### 3.1. Mechanisms of Cellulose Synthesis Regulation

Cellulose is the main component of the cell wall, and its content and structure directly affect agronomic traits such as the mechanical strength of rice stalks. At present, the envisioned process of the cellulose biosynthesis model is mainly divided into three steps (Figure 1) [34]: first, the membrane-associated sucrose synthase directs UDP-glucose to provide substrate for cellulose synthesis; three different types of cellulose synthase catalytic subunits (CESA) combine to form a cellulose synthase complex (CSC) with a six-petal rosette structure [30], which polymerizes glucose monomers into glucan chains while recycling the released UDP back to sucrose synthase; finally, the membrane-associated cellulase KORRIGAN (Kor) acts as a catalyst for the conversion of glucan chains into cellulose microfibrils, cutting defective glucan chains. The synthesis and assembly process of cellulose is complex and strictly regulated by genes coding for related enzymes and transcription factors.

#### 3.1.1. Cellulose Synthase

Cellulose synthase complexes are located on the cytoplasmic membrane and play an important role in cellulose synthesis. Under cryo-electron microscopy, the complex has a hexagonal wreath structure with a diameter of 25–30 nm. They use intracytoplasmic UDP-glucose as a substrate to stretch out the β-1,4-glucan chain through each rosette subunit, which, in turn, forms microfibrils on the outer side of the plasma membrane [35,36].

The rosette protein subunits of the cellulose synthase complex are encoded by the CESA gene [35,37]. Up to now, at least eleven CESAs genes have been successively identified in rice, three on chromosome 7, two on chromosome 3, and the other six on different rice chromosomes. Among them, *OsCesA1*, *OsCesA3*, and *OsCesA8* are closely co-expressed to form three different CESAs, which constitute the CSC involved in primary cell wall (PCW) synthesis, and *OsCesA4*, *OsCesA7*, and *OsCesA9* are closely co-expressed to form three different CESAs involved in the synthesis of secondary cell wall (SCW) [38]. Currently, many genes related to secondary cell wall cellulose synthase have been cloned and identified. Tanaka et al. screened three brittle culm mutants (*bc*) with mutated genes *OsCesA4*, *OsCesA7*, and *OsCesA9* using the Tos17 mutant library and found that all three had a significant decrease in stem cellulose content, indicating that the three genes are functionally not redundant [39]. The involvement of *CesA4* (*bc11*) in the synthesis of secondary cell walls was also verified by the use of the brittle stem mutant *bc11* in Zhou’s lab, in which the cellulose content of *bc11* decreased by 60% compared to the wild type, the mechanical strength of each tissue was significantly reduced, and the plants were short and stunted [40]. Wang et al. found that all the phenotypes of the S1-60 mutant were caused by recessive point mutations in the *OsCESA9* gene, which encodes cellulose synthase a subunit, and the stem cellulose content of the mutant decreases to 44.7% of the wild type, with a significant decrease in mechanical strength [41].

#### 3.1.2. Transcriptional Regulation of Cellulose Synthesis

Plant cell wall formation is a complex and delicate regulatory network that is regulated by many transcription factors. In the last decade, with the deepening of genetic and molecular biology research, a large number of genes involved in the transcriptional regulation of the plant secondary wall have been reported, and its regulatory network is mainly regulated by two major transcription factor families: NAC and MYB [42]. Among them, the NAC family of transcription factors is the top regulator of secondary wall biosynthesis, and overexpression of NAC genes causes ectopic deposition of the secondary wall, whereas suppressed expression will cause a decrease in the thickness of the secondary wall [43]. Specifically, these include *NAC SECONDARY-WALL-THICKENING-PROMOTING FACTOR* (*NST1*), *NST2*, *NST3*/*SND1*(*SECONDARY-WALL-ASSOCIATED NAC DOMAIN PROTEIN1*), *VASCULAR-RELATED NAC-DOMAIN6*(*VND6*), and *VND7* [41]. The downstream genes of NAC family transcription factors include *SND2*, *SND3*, *MYB20*, *MYB42*, *MYB46*, *MYB52*, *MYB54*, *MYB58*, *MYB63*, *MYB83*, *MYB85*, and *MYBl03*.

However, there are not many reports on the transcription factors of cellulose synthesis in the secondary walls of rice. Among the reported transcription factors, MYB46 is directly involved in regulating the expression of secondary-wall-synthesis-related genes *CesA4*, *CesA7*, and *CesA9* [44]. MYB58/63 directly regulate the expression of *OsCESA7*, a gene that regulates the cellulose synthase of the secondary wall [45]. *OsCELl* encodes the protein OsMYBl03L, which regulates secondary wall cellulose synthesis by binding to the CESAs and BCl promoter regions. Its mutant *cef1* exhibits reduced cellulose content and a brittle culm phenotype [46]. Huang et al. found that when *OsMYB61* was overexpressed, the cellulose content was significantly increased, the rigid cell walls were thickened, and it was directly regulated by the transcription factor NAC29/31, which plays an important role in the biosynthesis of the secondary wall through the DELLA-NAC signaling level pathway [47]. Ye et al. used the promoter region of *OsMYB61* as bait for yeast monohybrid screening to obtain OsSND2, which is also a NAC-like transcription factor. This protein directly binds to the promoter region of *OsMYB61* in vivo to regulate its expression level, and at the same time, it also serves as a key transcriptional regulator to regulate the expression of other MYBs and cellulose synthesis [48]. Chen et al. found that OsbHLH002 and OSH1 synergistically regulate the expression of *OsMYB61*. The loss-of-function mutant of the bHLH transcription factor OsbHLH002/OsICE1 exhibits a lodging phenotype. In addition, SLR1, the rice direct homologue of OsKNAT7, and OsNAC31 can interact with OsbHLH002 and OSH1 and regulate their ability to bind OsMYB61 [49]. Zhang et al. reported that a rice CCCH zinc finger structural protein IIP4 interacted with NAC29/NAC31, an upstream regulator of secondary wall cellulose synthesis, to block the downstream regulatory pathway in plants and act as an inhibitor in secondary wall cellulose biosynthesis. But it can be transferred to the cytoplasm after phosphorylation, allowing the above inhibition to be lifted [50]. In addition, Cao et al. recently reported a cellulose synthase co-expressed kinase (CSK1), which negatively regulates secondary wall cellulose biosynthesis by phosphorylating VND6, a major transcription factor related to cellulose synthesis in the nucleus. Interestingly, CSK1 and VND6 were found to be involved in the abscisic-acid-mediated regulation of cell growth and cellulose deposition, which are important components in the mechanism of cellulose synthesis, providing new clues for the study of cellulose synthesis regulation [51].

#### 3.1.3. Other Regulation of Cellulose

In addition to cellulose synthesis, the mode of cellulose deposition and degree of crystallinity also affect the mechanical strength of rice stalks. Studies have shown that the carbohydrate binding module (CBM) of the BC1N-terminal domain of COBRA-like proteins can bind crystalline cellulose, affecting the thickness of secondary cell walls and the mechanical strength of stems by regulating the crystallinity of microfilaments [52]. Moreover, there are some stem strength mutants that affect stalk strength by altering the directional deposition of cellulose microfilaments, such as *bc12* [53]. The process of cellulose assembly is extremely complex, and the exact mechanism remains to be investigated.

### 3.2. Mechanisms of Lignin Synthesis Regulation

The lignin synthesis pathway in rice is mainly the phenylalanine pathway. PAL, C4L, 4CL, CCR, CCoAOMT, F5H, COMT, CAD, and other enzymes are involved in this pathway. PAL and 4CL are the key enzymes in this pathway (Figure 2). The lignin content of *OsPAL* RNAi plants decreased significantly and the lignin content in *OsPAL8*-overexpressed plants was significantly increased, indicating that *OsPALs* could positively regulate lignin biosynthesis and accumulation [54]. Gui et al. found that among the genes regulating 4CL, although *Os4CL4* had the highest gene expression in the stem, *Os4CL3* had the highest expression abundance, and *Os4CL5* followed. The general rice phenotype is determined by the number of proteins that regulate this trait. The high expression abundance of these two genes indicated that more mRNA was transcribed by them in the stem, which promoted the accumulation of lignin monomeric G and S, thickened the sclerenchyma cells near the epidermis, and affected the stem thickness and mechanical strength of rice [55,56]. MYB transcription factor plays an indispensable role in rice lignin synthesis. *OsMYB30* can bind to *Os4CL3* and *Os4CL5* and activate their promoters, and it can also promote the accumulation of lignin monomers G and S [56]. OsMYB30 transcription factor can promote lignin synthesis by directly up-regulating gene expression of *OsPAL6* and *OsPAL8*. There are also *OsMYB55* and *OsMYB110* transcription factor regulatory genes in the MYB family. Both of them and *OsMYB30* can induce the accumulation of ferulic acid, an important intermediate for lignin synthesis, and indirectly affect the final synthesis pathway of lignin. Lignin deposition is no less important to rice stem strength than biosynthesis, and *OsMYB36a*, *OsMYB36b*, and *OsMYB36c* are collectively referred to as *OsMYB36s*. They realize the cooperative regulation of lignin deposition by encoding corresponding proteins and regulate the compensation lignification and corking of the inner cortex. But the specific mechanism is still to be studied [57]. CAD is a key enzyme for the synthesis and polymerization of lignin monomer. Both SH5 and qSH1 proteins of the BELL homeobox family can bind to *CAD* and may affect the expression of *CAD*, which may be caused by two pathways, respectively; when OSH15 interacts with SH5 to form a dimer, it directly inhibits lignin synthesis [58]. *SNB* is a gene that can affect the deposition of lignin in the cell wall. Jiang et al. found in their study that the *SNB* allele *SSH1* positively regulates *SH5* and *qSH1*, which, in turn, affects the action of CAD. However, due to the different effects of the two pathways, the specific regulatory trend is still unclear [59]. *CAD8B* is an important gene-encoding CAD enzyme. The high expression of this gene can promote lignin synthesis. It was found that the BZR1-NAC028 transcription factor complex can activate the expression of the *CAD8B* gene and then promote the transcription synthesis of the CAD enzyme [58]. In addition to CAD, CCR also plays an important role in the synthesis and polymerization of lignin monomers. According to previous studies, the synthesis pathway of the CCR enzyme is closely related to NAC transcription factors. Both NAC5 and NAC055 can activate *OsCCR10* expression, thus promoting the synthesis of CCR [60]. Of course, the role of NAC transcription factors in lignin synthesis is not only reflected in the regulatory pathway of CCR; NAC45 and NAC17 can positively regulate the lignin synthesis process [61]. Among them, NAC17 transcription factor can up-regulate the expression of genes encoding 4CL, CAD, LAC, PAL, PRX, and other key enzymes in lignin synthesis, such as *OsPAL7*, *prx137*, *prx22*, *OsCCR29*, *OsCAD8D*, etc., so as to promote lignin synthesis [62].

### 3.3. Mechanisms of Hemicellulose Synthesis Regulation

Hemicellulose and cellulose have similar main chain structures, so all main chains except the xylan main chain are synthesized by cellulose-synthase-like (CSL) protein. The sequences of the CSL protein families vary widely, and there is some variability in function. CSLA family members are responsible for the synthesis of mannan backbone, CSLC family members are responsible for the synthesis of xyloglucan backbone [30], and CSLF and CSLH family members can synthesize mixed glucan, but the formation of two different glycosidic bonds by one enzyme is contrary to the specificity of the enzyme, which still needs to be further studied [63].

At present, the composition and structure of hemicellulose have been clearly elucidated, but the genetic regulation mechanism of hemicellulose synthesis has been relatively slow. Three genes related to hemicellulose synthesis have now been cloned in rice, of which CSLD2 and CSLD4 belong to the cellulose-synthase-like protein D (OsCSLD) subfamily, and CSLF6 belongs to the cellulose synthase-like protein F (OsCSLF) subfamily. Li et al. found that the *CSLD4* mainly affected the structure of the primary cell wall, and the plant height was reduced after the gene mutation, the structure of the stem and root tip primary cell wall was defective, and the content of xylose and cellulose in the stem cell wall was reduced [64]. The *CSLF6* was mainly involved in the synthesis of xylan, and the mutation of this gene caused a significant decrease in xylan content and a significant decrease in stem mechanical strength at seedling and maturity stages [65], whereas *OsCSLD2* was only involved in the genetic regulation of root hair formation [66].

## 4. Research Progress on the Genetic Regulation Mechanism of Mechanical Strength of Rice Stems

### 4.1. QTL Related to Mechanical Strength of Rice Stems

The mechanical strength of rice stalks is a typical quantitative shape controlled by multiple genes. Currently, numerous scholars have used different genetic populations to map a large number of QTLs related to the mechanical strength of stems. A DH (double-haploid) population of 116 lines, constructed by Mu Ping, was used as a material to detect seven QTLs affecting stem diameter, which were distributed on chromosomes 1, 2, 4, 5, 7, and 8 and contributed 6.13~31.34% to phenotypic variation [67]; Xiao et al. identified three QTLs related to stem thickness distributed on chromosomes 1, 3, and 6 by using recombinant inbred lines of Nipponbare and Kasalath, consisting of 98 lines [68]; Kashiwagi et al. identified five QTLs on chromosomes 1, 7, 8, and 12 in a segregating population constructed from a cross between Nipponbare and Kasalath [69]; Yang Yaolong et al. mapped four relevant QTLs on chromosomes 1, 2, 6, and 8 [70]; Zhou Lei used the thick-stalked rice Shuhui 498 (R498) with lodging resistance and the thinner-stalked rice Yihui 3551 (R3551) as materials and finely located the primary effector locus qPND1, which controls the diameter of the spike-neck node of the rice stalks, at the 183 Kb interval at the end of the short arm of chromosome 1 and obtained the candidate gene *OsCKX2* [71]; Zhu Xiaoping used the recombinant inbred line population constructed from Gang 46B/A232 to conduct QTL analysis of the internode diameter of the last two nodes and the internode diameter of the last three nodes of rice and found that qRSID2-1and qRTID2-3 were repeatedly detected and located within 100 kb; these loci are closely related to the mechanical strength of rice stalks [72].

Although a number of related QTLs have been identified, due to the large amount of work required to identify these traits and the susceptibility of related traits to environmental influences, the reproducibility of the QTLs is poor, and there are few related genes cloned based on the QTL mapping of stem-strength-related traits. Therefore, it is of great theoretical and practical importance to further explore the favorable natural variation associated with stem strength in natural populations.

### 4.2. Genes Related to Mechanical Strength of Rice Stalks

The mechanical strength of rice stalks is closely related to various cell wall components. Affecting the expression of genes related to cellulose, lignin, and hemicellulose synthesis can lead to variations in the mechanical strength of rice stems. Numerous studies have shown that cellulose content is significantly reduced in loss-of-function mutants of the cellulose synthase genes *OsCesA4*, *OsCesA7*, and *OsCesA9*, all of which exhibit brittle culms [40,47,73,74]. The genes *BC10*, *BC3*, *BC14*, *BC25*, *BC12*, *OsHB4*, *OsMYB103L*, *OsGH9B1*, and *DPH1* have now been found to be involved in cellulose synthesis [53,75,76,77,78,79,80,81,82]. For example, OsMYB103L directly binds to and regulates the expression of *CESA4*, *CESA7*, and *CESA9* promoters. Yang et al. found that overexpression of *OsMYB103L* resulted in curling of rice leaves, significantly higher expression levels of several cellulose synthase genes, and significantly higher cellulose content [46,80]. In addition, genes such as *OsFH5* and *OsPMEI28* are involved in the regulation of lignin synthesis [83,84], and genes such as *OsNPC1*, *OsUGE3*, and *CslF6* are involved in the hemicellulose synthesis pathway [65,85,86].

Impaired expression of phytohormone-related genes likewise leads to variations in stem mechanical strength. For example, *OsSLR1*, *OsSPY*, and others affect rice stem mechanical strength through gibberellin (GA)-mediated regulatory pathways [87,88]. *OsCKX11*, *OsCKX9*, and others affect rice brittleness through the regulation of cytokinins [89,90]. *PAY1*, *OsMED14_1*, and others affect rice brittleness through the regulation of auxin [91,92].

Up to now, 78 genes related to the mechanical strength of rice stalks have been published on the China Rice Database (https://www.ricedata.cn/) website and the Gramene website (https://www.gramene.org/), distributed on 12 chromosomes (Table 1). Most of the cloned genes are related to cellulose and lignin synthesis. The excavation of these regulatory genes lays a solid foundation for the study of the regulatory mechanism of stalk mechanical strength and provides key genetic resources for the subsequent breeding of new high-yielding rice varieties.

## 5. Breeding and Its Application Value

Currently, great progress has been made in improving rice plant shape by regulating the mechanical strength of rice stalks in production applications [17].

### 5.1. Breeding of Rice for Resistance to Lodging

Since the “Green Revolution”, dwarf genes have been widely used in the breeding of lodging-resistant rice varieties, effectively alleviating the problem of lodging in rice [3]. However, due to the low biomass accumulation of traditional dwarf varieties, it is difficult to further improve rice yield. The combination of molecular breeding technology and traditional breeding is the fundamental means of improve the lodging resistance of rice varieties. LIU et al. found that *OsTCP19* was able to achieve a balanced regulation of lignin and cellulose synthesis by controlling the expression levels of transcription factors OsMYB108 and OsMYB103L. When *OsTCP19* was suppressed, lignin synthesis increased and cellulose synthesis decreased, making rice stalks become more brittle, less tough, and prone to lodging. While promoting the expression of *OsTCP19*, cellulose synthesis was enhanced, making rice stalks become tougher and improving resistance to lodging [116]. The molecular regulatory modules excavated by the study provide new ideas of how to improve the mechanical properties and lodging resistance traits of stems, which is valuable for breeding high-quality lodging-resistant rice seeds in agricultural production. Zhang et al. obtained the rice *erf34* mutant in rice through gene editing and found that the plant was dwarfed, RMD expression level decreased, cellulose and lignin content decreased, and stem mechanical strength weakened [117]. This study is expected to be applied to the practice of molecular breeding for resistance to lodging, which is a guide for rice cultivation and production.

### 5.2. Breeding of Brittle Rice

Studies have shown that brittle straw rice is characterized by low cellulose content, high lignin content, and high brittleness. It can be used as a new type of feed resource to realize “One seed, two harvests”; that is, mature rice for human consumption and straw for animal consumption, which has good economic benefits [118]. The straw can be returned to the field through silage and animal feed, which can realize a more friendly green cycle. However, most of the straws of brittle straw rice are brittle due to the mutation of cell wall components, causing rice to lodge, so there are few successful examples of brittle straw mutants directly applied in production.

In recent years, considerable progress has been made in the breeding of brittle straw rice. The China Rice Research Institute (CRI) used the dominant brittle straw rice ZGBCR obtained from the triple-cross combination II-32B/Xieqing early B/Dular population, crossed and backcrossed with Zhong9B/Zhong9A, to produce the world’s first sterile line rice with Indo-water type: Zhongcui A [119]. The Hefei Institute of Physical Sciences of the Chinese Academy of Sciences obtained a semi-dominant rice brittle straw mutant *sdbc1* via heavy ion mutagenesis of Yangjing 113 and crossed it with Wuyunjing 7 to obtain the first approved brittle culm variety Kefuijing 7, realizing the leap from theory to application of rice brittle stalk gene resources excavated in Chinese laboratories [120]. Wu Yuejin’s team at the Chinese Academy of Sciences bred a “brittle but not falling” brittle culm rice variety, Kecuijing 1, using the ideal brittle straw gene *IBC*. It overcomes the problems of short size, fragility, ease of falling, and low yield associated with the brittle straw variety by increasing the straw thickness, which compensates for the risk of lodging brought about by the variation in the cell wall components. At the same time, the cellulose and lignin contents of the straw decreased by 11.9% and 16.6%, respectively, and the hemicellulose content increased by 15.7%, which has the advantages of easy crushing, easy degradation, high nutritional value, and high efficiency of straw returning to the field, realizing a major breakthrough in the production application of rice brittle straw [121]. On this basis, Mou et al. bred a “dual use of cereals and grasses” variety with high yield, stress resistance, and cold resistance, Songkejing 169, using excellent japonica rice from the cold areas of Heilongjiang Province as the chassis variety and Keyunjing 7 as a brittle culm gene donor, which has great economic benefits in agricultural production.

## 6. Summary and Prospects

Lodging seriously affects rice production and food security [122]. The lodging of rice is mainly related to the mechanical strength of the stem, and the lodging-resistance capability is directly proportional to the mechanical strength of the stem. A large number of studies have shown that the structural components of the cell wall are closely related to the mechanical strength of rice stems; both cellulose and lignin play a key role. Therefore, it is of great significance to analyze the regulatory mechanism of cellulose and lignin synthesis and to search related genes to enrich the genetic regulation network of stem mechanical strength. In addition, the synthesis of cellulose and lignin is regulated by many transcription factors, and their regulatory networks are mainly regulated by two major transcription factor families: NAC and MYB. Huang et al. found that gibberellin (GA) signaling in rice promotes the synthesis of secondary wall cellulose by inhibiting the interaction between SLR1 and NACs [47]. Phytohormone-mediated transcriptional regulatory networks are expected to become a new research hotspot.

With the development of molecular biology and genetics, many scholars have discovered and cloned a number of key genes affecting the mechanical strength of stems using brittle culm mutants as research materials. At present, 78 genes related to the mechanical strength of rice stems have been published, distributed on 12 chromosomes.

In addition, great progress has been made in the practical application of improving rice plant type by regulating the mechanical strength of rice stems [17]. In actual agricultural production, rice lodging has become the key to restricting rice yields, and most lodging-resistant rice is hard to break down. The research of the golden cut-off point in regulating the “brittleness” and “lodging” of stems and cultivating high-quality rice that is “brittle but not lodging” has become a hotspot in the current research landscape. Wu Yuejin’s team at the Hefei Institute of Physical Sciences has bred a “brittle but not falling” brittle culm rice variety, Kecuijing 1, which achieved a major breakthrough on the journey from laboratory resources to actual production and application [121].

In the future, it will be necessary to open up new ideas and make more attempts using the known relevant regulatory mechanisms to study the mechanical strength of rice stems. At present, the types of signal molecules, receptors, and signal transduction pathways of cell wall formation are still unclear. Therefore, on the basis of existing studies, further exploring the signal regulation pathway of cell wall components and improving the regulatory network of stem mechanical strength will greatly accelerate the breeding process of high-quality rice, break the breeding barriers, and promote the revitalization of the seed industry.

## Figures and Tables

**Figure 1 plants-13-01726-f001:**
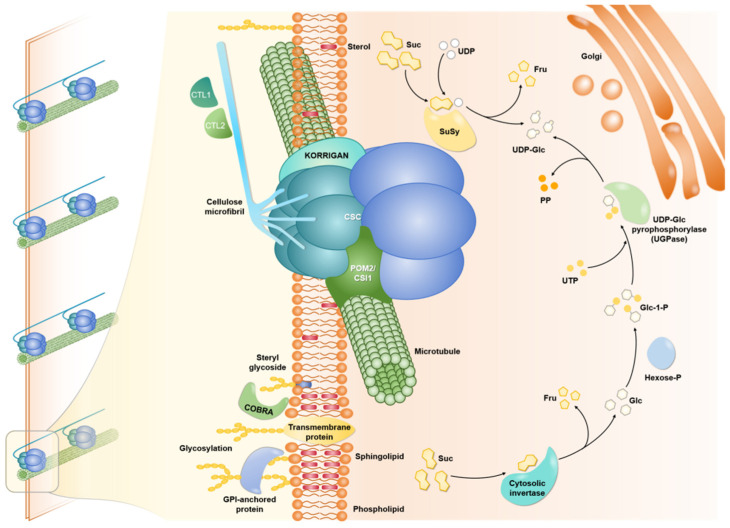
Cellulose synthesis model.

**Figure 2 plants-13-01726-f002:**
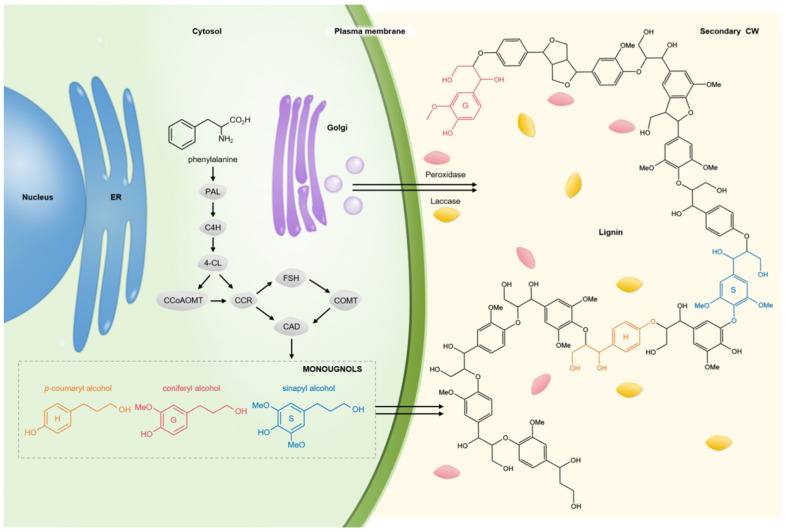
Lignin synthesis model.

**Table 1 plants-13-01726-t001:** Some of the genes related to the mechanical strength of rice stalks.

Target of Regulation	Gene	Phenotype	Localization of Chromosomes	Gene Function	Reference
Cellulose, lignin, and hemicelluloe	*OsCesA4*	Loss-of-function mutants exhibit brittle stems, dwarfed plant height, smaller leaves, thinner stalks, wilting of top leaves, and low fertility	1	*OsCesA4* encodes a cellulose synthase catalytic subunit and is involved in cellulose synthesis	Zhang et al., 2009; Huang et al., 2015 [40,47]
	*OsCesA7*	Loss-of-function mutants exhibit brittle culms and leaves, which break easily when bent; at the same time, the plants are dwarfed, the leaves and stems droop, and the number of tillers and fruiting rate are reduced	10	*OsCesA7* encodes a cellulose synthase catalytic subunit and is involved in cellulose synthesis	Huang et al., 2015; Wang et al., 2016 [47,73]
	*OsCesA9*	*OsCesA9* loss-of-function mutants have reduced plant height, smaller leaves, thinner stalks, wilted top leaves, and are partially sterile	9	*OsCesA9* encodes the catalytic subunit of cellulose synthase and is involved in cellulose synthesis	Li et al., 2017 [74]
	*BC10*	Loss-of-function mutants have reduced fibre content, increased leaf brittleness, shorter plant height at maturity, and reduced tiller number	5	*BC10* encodes a glycosyltransferase, which is required for rice cell wall biosynthesis	Zhou et al., 2010; Zhang et al., 2016 [75,93]
	*BC3*	Loss of *BC3* function results in plants showing slight dwarfing, shorter internodes, and reduced stem and leaf cellulose content	2	*BC3* encodes a classic promoter protein, OsDRP2B, which is involved in cellulose synthesis	Hirano et al., 2010; Xiong et al., 2010 [76,94]
	*BC14*	The cell wall cellulose content in *BC14* loss-of-function mutants decreased, the plants were short, the fertility decreased, and the seeds became smaller	2	*BC14* transports UDPG and regulates cellulose biosynthesis	Song et al., 2011; Zhang et al., 2011 [77,95]
	*BC25*	*BC25* loss-of-function mutants have reduced secondary cell wall thickness, reduced mechanical strength, and brittle internodes and leaf blades	3	*BC25* encodes a UDP glucuronide decarboxylase involved in cellulose synthesis, which is involved in sugar metabolism and affects cell wall synthesis	Xu et al., 2023 [78]
	*BC12*	*BC12* loss-of-function mutant plants are dwarfed with altered orientation of cellulose and cell wall components	9	*BC12* encodes a functional motor protein involved in the arrangement of microtubules during cell division	Zhang et al., 2010 [53]
	*OsHB4*	*OsHB4*-overexpressing plants produced narrow leaves curled towards the near-axis, and the angle of the leaves was reduced, resulting in an upright shape and a shorter plant height	3	OsHB4 binds to the promoters of *OsCAD2* and *OsCESA7*, repressing the expression of these two genes and regulating cell wall synthesis	Li et al., 2016; Zhang et al., 2018 [79,96]
	*OsMYB103L*	Overexpression of *OsMYB103L* led to curling of rice leaves, and the mechanical strength and cellulose content of *OsMYB103L* RNAi lines decreased	8	OsMYB103L directly binds to and regulates the expression of *CESA4*, *CESA7*, *CESA9*, and *BC1* promoters	Yang et al., 2014; Ye et al., 2015 [46,80]
	*OsGH9B1*	The cellulase activity of the transgenic lines increased significantly, and the mechanical strength of the stems decreased	2	*OsGH9B1* encodes the OsGH9B1 protein, which has endotype β-1,4-glucanase activity	Xie et al., 2013; Huang et al., 2019 [81,97]
	*DPH1*	Loss of *DPH1* function results in smaller tissues or organs, shorter roots and above-ground parts, a reduction in the number of primary and secondary branching peduncles, and a lower number of grains per spike	1	*DPH1* encodes OsSCD2 protein involved in lattice-protein-associated vesicular transport regulating cell expansion	Wang et al., 2022; Jiang et al., 2022 [82,98]
	*OsFH5*	Loss-of-gene-function mutants have reduced cellulose and lignin content, thinner secondary cell walls, and reduced mechanical strength of internodes	7	*OsFH5* encodes a formazin-like protein that regulates the correct spatial structure of actin-dynamic microtubule filaments and plays a key role in the morphology of rice	Zhang et al., 2011; Yang et al., 2011 [83,99]
	*OsPMEI28*	Loss-of-function mutants have 50% lower plant height, lower stalk diameter, lower tiller number, and delayed spiking	8	OsPMEI28 functions as a key component in the regulation of pectin methyl esterification levels	Nguyen et al., 2017 [84]
	*OsNPC1*	*OsNPC1* loss-of-function plants have brittle stem nodes that bend easily, brittle spike tips, and reduced thickness of thick-walled cells in stem nodes	3	NPC1 mediates the distribution of silicon and the deposition of secondary cell walls, affecting mechanical strength	Cao et al., 2016 [85]
	*OsUGE3*	*OsUGE3* loss-of-function plants showed significant decreases in plant height, biomass, spike weight, and thousand-grain weight, and the mechanical strength decreased	9	*OsUGE3* positively regulates cellulose and hemicellulose biosynthesis and increases polysaccharide deposition	Tang et al., 2022 [86]
	*CslF6*	The height and stem diameter of the *cslf6* knockout mutant decreased slightly, but the growth was normal during vegetative development	8	*CslF6* mediates the biosynthesis of MLG and affects MLG deposition, cell wall mechanical properties, and defensive response in rice vegetative tissues	Vega-Sánchez et al., 2012 [65]
Plant hormones	*OsSLR1*	*OsSLR1* loss-of-function mutants exhibit stalk tenderness, basal internode elongation, shorter and fewer root lengths, and increased cell length in the apical portion of the second leaf sheath	3	*SLR1* is a negative regulator of GA signaling and prevents downward GA signaling	Ikeda et al., 2001; Vleesschauwer et al., 2016 [87,100]
	*OsSPY*	*OsSPY* loss-of-function mutants exhibit a BR-deficient phenotype during the nutrient growth phase and an elongated phenotype with an excess of GA during the reproductive growth phase	8	OsSPY activates the expression of DELLA protein SLR1, a negative regulator of the gibberellin signaling pathway	Phanchaisri et al., 2012; Yano et al., 2019 [88,101]
	*RGA1*	*RGA1* loss-of-function mutant plants are dwarfed, with thickened stems, short and broad leaves, dense internodes, and low sensitivity to drought stress	5	*RGA1* encodes the α-subunit of GTP-binding proteins and affects the G-protein-dependent GA signaling pathway	Ueguchi-Tanaka et al., 2000; Ferrero-Serrano et al., 2016 [102,103]
	*DEP3*	Overexpression of *DEP3* causes a semi-dwarf phenotype, as well as a reduction in the length of rice stems, roots, leaves, seeds, and spikes	6	*DEP3* is involved in the expression of the growth inhibitor SLENDER1 in the gibberellin signaling pathway	Qiao et al., 2011; Liu et al., 2015 [104,105]
	OsNAC2	Loss-of-gene function mutants have increased tiller number, increased tiller angle, shorter plant height, and thinner stems	4	*OsNAC2* encodes the transcription factor NAC2, which binds to the promoters of GA-synthesis-related genes and represses their expression; it also up-regulates IAA-inactivation-related genes and down-regulates the expression of IAA-signaling-related genes and CK oxidation genes	Mao et al., 2007; Mao et al., 2020 [106,107]
	*OsCKX11*	The *OsCKX11* loss-of-function mutant showed a significant increase in the length and width of the apical 3 leaves, a thickening of the basal internode, an increase in the number of glumes per spike, a decrease in thousand-grain weight, and an increase in yield per plant	8	*OsCKX11* encodes a cytokinin oxidase that catalyzes the degradation of various cytokinins	Zhang et al., 2021; Rong et al., 2022 [89,108]
	*OsCKX9*	Loss-of-function mutants showed reduced plant height, increased tiller number, and reduced spike length, primary peduncle number, secondary peduncle number, and number of grains per spike	5	*OsCKX9* encodes a cytokinin oxidase that catalyzes the degradation of various cytokinins	Duan et al., 2019; Rong et al., 2022 [89,90]
	*OsCKX2*	The apical 3 leaves of the *OsCKX2* loss-of-function mutant appeared to be longer and wider, with a thickened basal internode and an increased thousand-grain weight	1	The *OsCKX2* gene encodes an enzyme that degrades cytokinin	Tu et al., 2021; Rong et al., 2022 [89,109]
	*PAY1*	The *PAY1* mutant showed increased plant height, reduced tiller number, reduced tiller angle, thickened stalks, larger spikes, longer internodes, increased spike branching, increased number of grains per spike, and increased yield	8	PAY1 improves rice plant size by affecting auxin polar transport and altering endogenous indole-3-acetic acid distribution	Zhao et al., 2015 [91]
	*OsMED14_1*	Loss-of-function mutants have reduced plant height, narrower leaves and stalks, reduced vascular system, fewer lateral root meristems, reduced spike branching and fruiting, and smaller seeds	8	OsMED14_1 physically interacts with the transcription factors YABBY5, TDR, and MADS29 and may regulate auxin homeostasis, ultimately affecting rice lateral organ/leaf, microspore, and seed development	Malik et al., 2020 [92]
	*OsGRF7*	*OsGRF7* overexpressing lines were compact and semi-dwarf, with increased stem wall thickness and narrowed leaf angle	12	OsGRF7 binds to the promoter regions of the cytochrome P450 gene *OsCYP714B1* and the growth factor response factor gene *OsARF12* and is involved in gibberellin synthesis and auxin signaling	Chen et al., 2020; Chen et al., 2020 [110,111]
	*OsSPL17*	The root elongation response of the *SPL17* knockout mutant is insensitive to NO_3_^−^ and rac-GR24 application	9	*SPL17* promote or maintain the formation of reproductive growth by terminating/inhibiting vegetative growth and have an inhibitory effect on panicle bract formation	Wang et al., 2021; Sun et al., 2021 [112,113]
	*OsSPL14*	The loss of *OsSPL14* function led to a decrease in tillering and an increase in grain number per spike and thousand-grain weight, while the stem became thicker and the lodging resistance was enhanced	8	The transcription factor OsSPL14 binds to the promoter region of auxin export vectors to activate their expression, thereby regulating auxin trafficking and distribution	Jiao et al., 2010; Li et al., 2022 [114,115]

## Data Availability

Data availability is not applicable to this article as no new data were created or analyzed in this study.

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
