# Peer review of "Research Progress on Mechanical Strength of Rice Stalks"

_plants, 2024, doi:10.3390/plants13131726_

Round 1

Reviewer 1 Report

Comments and Suggestions for Authors

The manuscript reviews the progress of research on the mechanical strength of rice stalks, for potential application to enhance lodging resistance in rice. Overall, the manuscript reads well. However, I think that Section 4.2 of the manuscript was not thoroughly discussed. The manuscript would be more informative if this section is improved. Below are some specific comments:

Section 4.2. Genes related to mechanical strength of rice stalks can be improved by providing further information regarding research/application (if any) on using those genes (in Table 1) for improvement of lodging resistance in rice, especially using genetic engineering and/or genome editing approach. In the current manuscript, section 4.2 only describes the works that have been done so far without a thorough analysis.

The Phenotype column in Table 1 is very hard to follow. It did not explain whether these phenotypes are from plants with loss function or overexpressed the gene of interest. Most of the information in the column were presented continuously without separation between genes.

Author Response

Please see in attachment.

Reviewer 2 Report

Comments and Suggestions for Authors

 A more detailed description of the measurement procedure.

Author Response

Please see in attachment.

Reviewer 3 Report

Comments and Suggestions for Authors

The authors presented the review article describing one of the most important factors restricting rice production, the lodging. In this nicely written manuscript, the authors drew attention to the lodging resistance of rice, which directly can be relate to the mechanical strength of the stalks. The authors reviewed the cell wall structure, components and its genetic regulatory mechanism. At the end of this review article, the authors present the new progress in genetic breeding, and further indicated the specific scientific problems that need to be solve in the future, in order to provide the improvement of rice breeding.

All of mine detailed corrections are incorporate directly in the text manuscript. There are many mistakes in the text manuscript that should be correct and some sentences should be re-written. Also, in the whole manuscript, seems to be that references are quite mess up and really are confusing so the detailed check is need.

Author Response

Please see in attachment.

Reviewer 4 Report

Comments and Suggestions for Authors

The manuscript is well organized and presented. The references are adequate and permanent, but certain reference was missing.

Also one line 21, the scientific name should be in italics.

The MS is within the scope of the journal. On a scale of 1 to 10 with 10 as the highest, I would give the MS a rating of 7 and recommend its acceptance for publication. It should be of interest to the scientific community.

Author Response

Please see in attachment.

Round 2

Reviewer 3 Report

Comments and Suggestions for Authors

The authors applied all of mine suggestions and recommendations so, in my opinion, the article now can be accepted for publication.